# Assessment of Foot Strike Angle and Forward Propulsion with Wearable Sensors in People with Stroke

**DOI:** 10.3390/s24020710

**Published:** 2024-01-22

**Authors:** Carmen J. Ensink, Cheriel Hofstad, Theo Theunissen, Noël L. W. Keijsers

**Affiliations:** 1Department of Research, Sint Maartenskliniek, 6500 GM Nijmegen, The Netherlands; 2Department of Sensorimotor Neuroscience, Donders Institute for Brain, Cognition and Behaviour, Radboud University, 6500 HB Nijmegen, The Netherlands; 3Department of Information and Communication Technology, HAN University of Applied Sciences, 6524 RN Nijmegen, The Netherlands; 4Department of Rehabilitation, Donders Institute for Brain, Cognition and Behaviour, Radboud University Medical Center, 6500 HB Nijmegen, The Netherlands

**Keywords:** foot strike angle, gait, inertial measurement unit, wearable sensor

## Abstract

Effective retraining of foot elevation and forward propulsion is a critical aspect of gait rehabilitation therapy after stroke, but valuable feedback to enhance these functions is often absent during home-based training. To enable feedback at home, this study assesses the validity of an inertial measurement unit (IMU) to measure the foot strike angle (FSA), and explores eight different kinematic parameters as potential indicators for forward propulsion. Twelve people with stroke performed walking trials while equipped with five IMUs and markers for optical motion analysis (the gold standard). The validity of the IMU-based FSA was assessed via Bland–Altman analysis, ICC, and the repeatability coefficient. Eight different kinematic parameters were compared to the forward propulsion via Pearson correlation. Analyses were performed on a stride-by-stride level and within-subject level. On a stride-by-stride level, the mean difference between the IMU-based FSA and OMCS-based FSA was 1.4 (95% confidence: −3.0; 5.9) degrees, with ICC = 0.97, and a repeatability coefficient of 5.3 degrees. The mean difference for the within-subject analysis was 1.5 (95% confidence: −1.0; 3.9) degrees, with a mean repeatability coefficient of 3.1 (SD: 2.0) degrees. Pearson’s r value for all the studied parameters with forward propulsion were below 0.75 for the within-subject analysis, while on a stride-by-stride level the foot angle upon terminal contact and maximum foot angular velocity could be indicative for the peak forward propulsion. In conclusion, the FSA can accurately be assessed with an IMU on the foot in people with stroke during regular walking. However, no suitable kinematic indicator for forward propulsion was identified based on foot and shank movement that could be used for feedback in people with stroke.

## 1. Introduction

Stroke survivors commonly face challenges related to impaired balance and gait, often attributed to diminished foot elevation and inadequate forward propulsion [1]. These challenges significantly increase the risk of falls and result in decreased gait speed [2,3], negatively impacting daily activities and overall quality of life [4]. Therefore, effectively retraining foot elevation and forward propulsion is a critical aspect of gait rehabilitation therapy [5]. During in-clinic therapy, therapists provide valuable feedback to patients to enhance these functions to further improve their gait pattern. Given that stroke survivors commonly experience not only motor impairments but also sensory deficits [6], this feedback is of utmost importance for successful rehabilitation. However, once patients are discharged from clinical care, they no longer receive feedback on their gait pattern during home-based training.

One potential solution is to integrate inertial measurement units (IMUs) for real-time feedback within home-based training. Since reduced foot elevation and insufficient forward propulsion are major factors contributing to gait problems in stroke [7,8], outcome parameters for feedback assessed with IMUs should be related to these impairments. Reduced foot elevation often results from weakness in the ankle dorsiflexors and is often characterized by toe landing rather than heel strike [7]. Therefore, the ankle angle or foot strike pattern (forefoot, midfoot, or rearfoot) could be used to train foot elevation. Previous research has demonstrated that IMUs can accurately estimate lower limb kinematics and spatiotemporal parameters [9,10,11,12]. Although the insights offered by lower limb joint angles are valuable [7,8], at least two sensors are needed to measure the angles of one joint, one on the proximal and one on the distal segment [11]. On the other hand, previous research on running kinematics revealed that a single IMU on the foot was able to distinguish between foot strike patterns (forefoot, midfoot, and rearfoot) [13,14]. Therefore, IMUs have the potential to offer valuable feedback to people with stroke on the foot strike angle (FSA), the angle formed between the foot and the walking surface upon initial contact (IC).

Besides feedback on the FSA, feedback on forward propulsion could also be useful for stroke survivors during exercise performance at home. However, IMUs cannot measure force directly, making the quantification of forward propulsion challenging through this modality [15]. Therefore, it is interesting to study if there are indicative gait characteristics for forward propulsion that can be measured with an IMU. It is generally thought that increasing forward propulsion leads to a higher gait speed with larger strides, resulting in altered kinematics of the foot and lower leg such as an increased angular velocity of the foot and a larger shank-to-vertical angle upon terminal contact (TC) [7,8,12,16,17,18]. Therefore, changes in foot and shank kinematics might be indicative of the generated forward propulsion. Pieper et al. [12] found support for this idea via a strong correlation between peak shank acceleration and peak forward propulsion in healthy individuals, both at individual and group levels. Although Pieper et al. mimicked pathological gait patterns by imposing unilateral movement constraints on the ankle and knee joint, it is unknown if the correlation holds true in pathological gait (e.g., stroke survivors).

The present study has two objectives: (1) to validate the accuracy of the IMU-derived FSA in individuals with stroke against the gold-standard optical motion capture system (OMCS), and (2) to identify IMU-derived parameters that are indicative of forward propulsion in individuals with stroke. We hypothesized that the FSA could be measured with high accuracy (a deviation from the gold standard of <5 degrees), based on previous work regarding the shank angle, which reached a mean difference of 0.7 degrees with a repeatability coefficient of 4.2 degrees compared to that of the OMCS [10]. Regarding the second aim, we anticipated that several foot and shank kinematic variables during the gait cycle would exhibit a moderate correlation (Pearson correlation coefficients ranging from 0.5 to 0.75) with forward propulsion. Based on the general belief expressed in the literature that decreased forward propulsion leads to altered gait kinematics, decreased gait speed, and shorter stride lengths [7,8,12,16,17,18,19], we measured the foot and shank angle upon TC, the maximum angular velocity and angular acceleration during the stance phase (IC to TC) of both the foot and shank, the maximum shank linear acceleration, and the stride length with the gold standard (OMCS), and evaluated these parameters as indicators for the actual forward propulsion. These parameters were chosen based on the previously found promising results for the shank linear acceleration [12], gait speed [12,18], stride length [17,18], and peak angular velocity of the lower limb segments [18,19], and their potential to be derived from only a single IMU. Finally, the same metrics were calculated with the IMU system to verify that the IMU system reaches similar correlations between these metrics and the forward propulsion.

## 2. Materials and Methods

### 2.1. Participants

Twelve participants were recruited between January 2023 and June 2023 from physiotherapy practices in and around Nijmegen, as well as from social media groups for stroke survivors. Participants were eligible when they had experienced a stroke at least 6 months prior, were at least 18 years old, had unilateral motor deficits, and could walk for at least 5 min without assistive devices. Individuals were excluded if they lacked a sufficient cognitive ability to understand basic instructions, had a history of orthopedic or neurologic disorders (excluding stroke) that could affect gait or balance, had undergone surgery to correct drop foot, or were unable to perform any ankle flexion–extension. All participants gave their written informed consent prior to participation.

The study protocol was in line with the Declaration of Helsinki and was granted an exemption by the Dutch Medical Scientific Research Act (WMO) from ‘METC Oost-Nederland’ (identification number: 2021-13295).

### 2.2. Materials

Participants were equipped with five IMUs (MTw Awinda, Movella, Enschede, The Netherlands) attached to the dorsal side of both feet, the anterior aspect of their shanks, and the lower back (L4/5), along with 20 reflective markers for the OMCS. Reflective markers were placed according to the VICON plug-and-gait lower body model [20]. MT Manager software suite version 2019.2 was used for the data capture of the IMUs. Participants walked on the GRAIL (Gait Real-time Interactive Analysis Lab, (Motek Medical, Amsterdam, The Netherlands)), an instrumented treadmill with an eight-camera OMCS (VICON, Oxford, UK), embedded force plates (Motek Medical, Amsterdam, The Netherlands), and a wide (180°) circular screen in front of the treadmill, creating a virtual environment. The IMU and OMCS both recorded at a sample frequency of 100 Hz, while the force plates operated at 1000 Hz. All systems were time-synchronized by a high–low pulse, with the OMCS serving as master.

### 2.3. Measurements

After a familiarization period, participants performed five walking trials on the GRAIL. The first and last trials involved self-paced regular walking, where participants had control over the speed of the treadmill by positioning themselves at the front (to accelerate) or at the back (to decelerate) of the belt [21]. Data were captured for two minutes starting when participants indicated that they were at a comfortable walking speed. Trials two to four introduced variability in the FSA and anterior–posterior propulsion by providing feedback on either their FSA, propulsion, or both simultaneously. Feedback was provided visually via a vertical slide bar on the GRAIL’s screen, with the slide moving upwards to the green end or downwards to the red end based on the participant’s performance. The second and third trials were randomized across subjects with feedback on either the FSA (based on OMCS data) or propulsion (based on the force plate data). During the fourth trial, participants received feedback on both parameters. At the start of each feedback trial, participants walked 10 strides without feedback. The GRAIL system calculated their regular FSA and propulsion, followed by 2 min of walking with feedback, during which data were captured. All measurements and visual feedback were embedded in a custom-built GRAIL application.

### 2.4. Data Processing

IMU data captured by MT Manager software (2019.2) included angular velocity and acceleration data in the sensor frame, acceleration in the earth frame, and orientation in a quaternion and Euler angle format. OMCS data were captured by VICON Nexus software (version 2.4). All further data processing and analyses were performed in Python 3.10.

A second-order low-pass Butterworth filter was applied to the angular velocity (cut-off frequency of 15 Hz) and acceleration data (cut-off frequency of 17 Hz) of the IMUs [22,23]. OMCS data were similarly filtered using a second-order low-pass Butterworth filter with a 15 Hz cut-off frequency. Force plate data were filtered using a fourth-order low-pass Butterworth filter with a 20 Hz cut-off frequency [24].

All of the code for data processing and analysis is available at the following link: https://github.com/SintMaartenskliniek/MovingReality (Release: “Validation study”, tag: “v1.0.0”, accessed on 6 January 2024).

### 2.5. Data Analysis

Each trial had a data recording time of 120 s. Data recording started 10 s after initiating the trial to exclude the initial acceleration phase to reach the comfortable walking speed. Data recording was stopped before the participant began decelerating to end the trial.

For the OMCS data, gait events were determined based on the validated method of Zeni et al. [24]. This method identifies IC as the instant when the velocity vector in the anterior–posterior direction of the heel marker crosses zero in the posterior direction. TC corresponds to the instant where the velocity vector in the anterior–posterior direction of the toe marker crosses zero in the anterior direction. For IMU data, IC events were identified at the instant of the first zero-crossing of the angular velocity around the mediolateral axis after mid-swing (maximum angular velocity around the mediolateral axis) [23]. TC events were identified at the peak vertical acceleration between mid-swing events (maximum angular velocity around the mediolateral axis) [23]. The foot flat phase, when the foot was flat on the walking surface, was identified between TC and the mid-swing of the contralateral side.

The OMCS global coordinate system was defined with the *z*-axis aligned to the vertical direction, the *y*-axis aligned to the walking direction, and the *x*-axis perpendicular to this plane. The IMUs used in this study also provide acceleration in the global frame. The IMU global frame is defined such that the *x*-axis is pointing to the magnetic north, the *z*-axis is aligned with the gravity direction, and the *y*-axis is perpendicular to this plane. Figure 1 shows a schematic illustration of the experimental setup.

The foot segment was defined between the position of the toe and heel markers from the OMCS data (Equation (1)), after which the foot angle during the gait cycle was calculated in accordance with Equation (2). During the foot flat phase, the foot angle was considered to be zero degrees. Therefore, the foot angle was adjusted by subtracting the mean foot angle measured during the mid-stance of the first 10 strides (Equation (2)). Subsequently, the foot angle was converted from radians into degrees in accordance with Equation (3).
Foot segment _OMCS_ = position _TOE MARKER_ − position _HEEL MARKER_,(1)
(2)$$\mathrm{Foot}\;\mathrm{angle}{\;}_{\mathrm{OMCS}}=\tan^{-1}(\frac{{foot\;segment}_{\;OMCS\;vertical\;component}}{{foot\;segment}_{\;OMCS\;walking\;direction\;component}}),$$
Foot angle _OMCS_ = (foot angle _OMCS_ − mean (foot angle _OMCS mid-stance of stride 1 to 10_)) × 180/π,(3)

Finally, the foot strike angle was determined for each IC event based on the OMCS event algorithm (Equation (4)):Foot strike angle _OMCS, IMU_ = foot angle _OMCS, IMU at IC_,(4)

For IMU data, the Euler angles directly retrieved from the sensor were used as the estimated foot angles, with the Euler pitch angle corresponding to the foot angle of interest. Importantly, we assumed that the sensor axes were aligned with the axes of the foot segment. The foot angle as measured with the IMU is tilted due to attachment to the dorsal side of the foot (see Figure 2). This was corrected by subtracting the mean foot angle measured during the foot flat phase of the first 10 strides in accordance with Equation (5), considering the foot angle during the foot flat phase to be zero degrees. Finally, the foot strike angle was determined as the foot angle upon IC, for each IC event based on the IMU event algorithm (Equation (4)).
Foot angle _IMU_ = (foot angle _IMU_ − mean (foot angle _IMU foot flat of stride 1 to 10_)),(5)

For our second aim, the parameter of interest was forward propulsion. In the literature, two main approaches have been used to quantify this parameter. First, forward propulsion has been defined as the area under the curve (AUC) of the measured anterior–posterior ground reaction force (GRF) during each push-off [25]. This involves the numerical integration of the GRF in the anterior–posterior direction from the breaking-to-propulsion transition until TC is observed with bodyweight normalization (Equation (6) and Figure 3) [25]. Second, forward propulsion has been defined as the maximum value of the anterior–posterior GRF during each push-off (Equation (7)).
(6)$$\mathrm{Forward}\;\mathrm{propulsion}{\;}_{\mathrm{AUC}}=\int_{BPT}^{TC}{GRF}_{AP\;direction}dt,\;\mathrm{with}\;\mathrm{dt}=1/\mathrm{sample}\;\mathrm{frequency},\;\mathrm{TC}\;=\;\mathrm{terminal}\;\mathrm{contact},\;\;\mathrm{BPT}\;=\;\mathrm{breaking-to-propulsion}\;\mathrm{transition},\;\mathrm{GRF}\;=\;\mathrm{ground reaction force},\;\;\mathrm{and AP}\;=\;\mathrm{anterior-posterior},$$
(7)$$\mathrm{Forward}\;\mathrm{propulsion}{\;}_{\mathrm{peak}}=\mathrm{maximum}\;({GRF}_{AP\;direction}),\;{\mathrm{with GRF}}_{\mathrm{AP direction}}\;\mathrm{for each breaking-to-propulsion transition until terminal contact}$$

Eight parameters were identified as possible indicators for forward propulsion: the foot and shank angle upon TC, the maximum angular velocity and angular acceleration during the stance phase (IC to TC) of both the foot and shank, maximum shank linear acceleration, and the stride length. The calculation of the foot angle over time is described above for both systems. For each gait cycle, the foot angle upon TC was calculated. For OMCS data, the shank angle over time was calculated in accordance with Equations (8) and (9), while the IMU-based shank angle was directly derived from the Euler angle of the sensor output. Again, the shank angle upon TC for both systems was calculated for each gait cycle.
Shank segment _OMCS_ = position _KNEE MARKER_ − position _ANKLE MARKER_,(8)
(9)$$\mathrm{Shank}\;\mathrm{angle}{\;}_{\mathrm{OMCS}}=\tan^{-1}\left(\frac{{shank\;segment}_{\;OMCS\;vertical\;component}}{{shank\;segment}_{\;OMCS\;walking\;direction\;component}}\right)$$

The foot and shank angular velocity were calculated as the first derivative of the foot and shank angle for the OMCS, respectively. For the IMU-based foot and shank angular velocity, the angular velocity directly measured from the gyroscope was used. The foot and shank angular acceleration were subsequently calculated as the derivative of the foot and shank angular velocity for both measurement systems. Finally, the maximum value of each of the parameters for each gait cycle was taken.

The linear acceleration of the shank was calculated as the square root of the squared acceleration in the global frame in the horizontal plane (Equation (10)) for both systems. For the OMCS, the acceleration along the *x*- and *y*-axis was calculated with the second derivative of the x- and y-positions of the shank segment defined in Equation (8). For the IMUs, the acceleration in the global frame was directly retrieved from the IMU data. For the shank’s linear acceleration, again, the maximum value during each stance phase was computed.
(10)$$\mathrm{Shank}\;\mathrm{linear}\;\mathrm{acceleration}=\sqrt{{{(acceleration}_{x-axis})}^{2}+{{(acceleration}_{y-axis})}^{2}}$$

### 2.6. Statistical Analysis

Participant characteristics were reported using descriptive statistics. The normality of the data was tested using the Shapiro–Wilk test, and results were reported accordingly. To assess the reliability and agreement of the IMU-derived FSA compared to those of the gold standard, intraclass correlation and Bland–Altman analysis were performed for all strides of all participants, as well as for each participant individually. The latter, referred to as within-subject analysis, was performed to evaluate whether or not the parameters could be used as feedback for individualized home-based training. To determine if a potential parameter was a suitable indicator for forward propulsion, the Pearson correlation coefficients between the potential parameters (foot angle at TC, shank angle at TC, maximum foot angular velocity, maximum shank angular velocity, maximum foot angular acceleration, maximum shank angular acceleration, maximum shank linear acceleration, and stride length), the AUC and peak forward propulsion were calculated. This analysis was performed for both the OMCS and IMU system. This dual approach allowed us to evaluate the potential of these parameters to serve as indicators for forward propulsion (AUC and peak) based on the gold-standard method OMCS, and to confirm the IMU’s ability to serve the same purpose. Both ICC and Pearson correlation values were interpreted as weak (<0.5), moderate (0.5–0.75), good (0.75–0.9), and excellent (>0.9) reliability and correlation [26]. A parameter was considered a possible indicator for forward propulsion if the significant (*p* < 0.05) Pearson correlation value was at least good (r > 0.75).

## 3. Results

### 3.1. Participant Characteristics

All 12 participants (7 male/5 female) were previously enrolled in a gait rehabilitation training program post-stroke. Their mean age was 61 years (SD: 9.5) with a median time since stroke onset of 25 months (6 to 210 months). Eight participants experienced an ischemic stroke, two experienced a hemorrhagic stroke, and from two participants the type of stroke was unknown. The average comfortable gait speed was 1.0 (SD: 0.3) m/s. Participant characteristics are presented in Table 1.

### 3.2. Foot Strike Angle Validation

In total, 11,985 strides from all trials and all participants were included for stride-by-stride validity analysis. Excellent reliability of the IMU-based FSA compared to the OMCS-based FSA was found via the ICC (ICC (3, 1) = 0.97, 95%CI: [0.96; 0.97]). Figure 4 shows the Bland–Altman analysis of the FSA measured on a stride-by-stride basis. Differences between the IMU-based FSA and OMCS-based FSA were on average 1.4 degrees, with 95% limits of agreement ranging from −3.0 to 5.9 degrees. The repeatability coefficient was 5.3 degrees.

For the within-subject analysis, the step count per subject ranged from 630 to 1283 steps. Differences between the IMU-based and OMCS-based FSA were on average 1.5 degrees, with 95% limits of agreement ranging from −1.0 to 3.9 degrees (Figure 5). The mean repeatability coefficient for the within-subject analysis was 3.1 (SD: 2.0) degrees. Figure A1 in Appendix A shows the Bland–Altman analysis of the FSA on a stride-by-stride level for each participant.

### 3.3. Indicative Parameter for Forward Propulsion

Out of the 11,985 strides recorded in total, 7591 strides were suitable for a further analysis of propulsive force, as they involved only one foot on a single force plate. For each individual, between 1693 and 931 strides were included in this analysis (median 665 strides).

All IMU-based indicators for forward propulsion demonstrated only weak to moderate Pearson correlation coefficients with the AUC forward propulsion on a stride-by-stride level (see Table 2). The equivalent OMCS-based parameters revealed similar weak to moderate Pearson correlation coefficients. The mean and SD of the Pearson correlation between the indicators for forward propulsion and the measured AUC forward propulsion for the within-subject analysis are presented in Table 3. The mean Pearson correlations ranged between 0.06 and 0.63 with relatively high SD values, indicating large differences between subjects. Appendix A, Figure A2, includes correlation graphs of each of the parameters with the AUC forward propulsion.

All IMU-based indicators for the peak forward propulsion demonstrated only weak to moderate Pearson correlation coefficients in the stride-by-stride analysis, except for stride length (r = 0.76) (see Table 4). The equivalent OMCS-based parameters revealed higher Pearson correlation coefficients of up to r = 0.77 for the maximum foot angular velocity and r = 0.76 for the foot angle upon TC. The mean and SD of the Pearson correlation between the indicators for forward propulsion and the measured peak forward propulsion for the within-subject analysis are presented in Table 5. While the mean Pearson correlation for the within-subject analysis did not exceed ‘moderate’ correlation values, the relatively high SD values between 0.19 and 0.49 indicate large differences between subjects. Appendix A, Figure A3, includes correlation graphs of each of the parameters with the peak forward propulsion.

## 4. Discussion

The present study aimed to evaluate the accuracy of the IMU-derived FSA and to identify IMU-derived indicators for forward propulsion in individuals with stroke. The results show high accuracy for the IMU-derived FSA compared to that of the gold standard. Regarding the second aim, weak to moderate correlations between eight potential indicators and the measured forward propulsion were found.

The stride-by-stride evaluation revealed a mean difference of 1.4 degrees with a standard deviation of 2.3 degrees for the IMU-derived FSA, coupled with an excellent intraclass correlation (>0.9) when compared to that of the gold standard, indicating an acceptable level of accuracy. Previous research on the assessment of FSA with IMUs was performed in healthy participants during running. Although running is inherently different from walking, our results surpassed the accuracy even when analyzed on a stride-by-stride basis (3.9 ± 5.3 degrees) [14]. Furthermore, the results of this study are in line with the accuracy of estimated shank angles in walking, both of which are based on the same principle of estimating segment orientation from a single IMU [10]. When the FSA was averaged across all strides within each participant, every participant had a difference of less than 5 degrees compared to that under the gold standard (see Figure 5). More importantly, while the repeatability coefficient on a stride-by-stride basis was just above 5 degrees (5.3), a mean repeatability coefficient of only 3.1 degrees was found when analyzed within subjects. Given that the repeatability within subjects is well within the set limit of 5 degrees and only slightly exceeds it in the stride-by-stride analysis, we conclude that the FSA could accurately be assessed with an IMU in people with stroke.

For the second aim, potential indicators for forward propulsion, defined as either the AUC or the peak anterior–posterior GRF, were evaluated. Based on the previous literature [7,8,12,16,17,18,19], seven kinematic parameters of the shank and foot, as well as stride length, were evaluated by calculating the correlation coefficient with the generated forward propulsion. The stride-by-stride analysis for AUC forward propulsion yielded weak to moderate correlations (see Table 2). When considering peak forward propulsion, previous research has shown that shank linear acceleration could serve as a good to excellent indicator [12]. Unfortunately, our study did not replicate this correlation for either the OMCS- (r = 0.35) or IMU-derived (r = 0.38) shank linear acceleration parameter (see Table 4). However, maximum foot angular velocity, foot angle upon TC, and stride length marginally exceeded the threshold for a good correlation, suggesting their potential as indicators for peak forward propulsion, aligning with the review of Roelker et al. [18]. Unfortunately, only the IMU-based equivalent correlation coefficient for stride length reached the level of a good correlation, whereas the maximum foot angular velocity and foot angle at TC had only a moderate correlation. The absence of strong correlations between any of the parameters with forward propulsion on a stride-by-stride basis might be attributed to heterogeneity in gait patterns within our study population. While all participants were chronic stroke patients with affected gait, there were notable differences in gait speed and gait pattern, including varying degrees of stiff knee gait and compensatory strategies such as hip circumduction. This altered gait in stroke patients could also explain the disparity between our study and the research of Pieper and colleagues [12], which involved healthy participants tested during regular walking and walking with simulated pathological gait. Based on the current study, we conclude that none of the proposed IMU-derived indicators could serve as a valid indicator for forward propulsion.

Since a general application of sensors is to integrate them in real-time home-based training settings [27,28], the individual participant correlation between the potential indicators and forward propulsion was also evaluated. Averaged across subjects, this within-subject analysis yielded moderate correlations for the AUC and peak forward propulsion. Again, the correlation coefficients of the OMCS-based parameters were lower than their IMU-based equivalent parameters. Importantly, substantial inter-individual variability in the various potential indicative parameters for both AUC and peak forward propulsion was found, as indicated by the high SDs across participants (see Table 3 and Table 5). Nevertheless, none of the explored parameters reached the minimum requirement of a ‘good’ correlation (r > 0.75) for a substantial number of individuals. Therefore, we do not consider any of the studied parameters as appropriate to provide feedback on forward propulsion to improve the gait pattern.

This study has some limitations. Firstly, the evaluation of straight-ahead treadmill walking, though common in research protocols, does not fully capture the complexity of real-life walking scenarios involving curved paths, uphill, downhill terrain, and uneven surfaces. Gait kinetics and kinematics can notably differ under these diverse conditions compared to those under straight-ahead walking [29]. Therefore, the ecological validity of our findings, both in terms of the validity of the FSA and indicators of forward propulsion in real-world walking scenarios, warrants further investigation. Secondly, the discrepancy found in correlations of the forward propulsion with possibly indicative parameters between OMCS-derived and IMU-derived parameters suggests that there is a difference between the parameters when obtained with the OMCS and IMU. Enhancing the validity of the IMU-based parameters would be valuable and could result in correlation values similar to the OMCS-based equivalents with forward propulsion. This would mean that maximum foot angular velocity and foot angle upon TC could be used to assess an individual’s peak forward propulsion based on multiple strides. Thirdly, our study population consisted of twelve participants based on the recommendation as a rule of thumb for pilot studies [30]. While this was a convenient sample to test the usability of a feedback system for the first time, this limited number of participants might not include all different variations of gait patterns. Future research could explore the effect of differences in gait patterns on the correlation of certain gait characteristics with forward propulsion. Lastly, our choice to evaluate relatively simple parameters as indicators for forward propulsion was driven by the potential application of a real-time feedback system for home-based rehabilitation. Prioritizing computational efficiency and usability, the number of required IMUs was limited to one or a maximum of two attached to the affected leg. Furthermore, other parameters that could be derived from the sensors, such as the timing of the selected parameters in the gait cycle, could also be valuable to estimate the forward propulsion. According to the literature [12,17], there was no reason to believe that the timing of the selected parameters was an indicator for forward propulsion. Nevertheless, the potential of these parameters and their combination should be explored in future studies. However, we acknowledge that individuals with stroke use diverse gait strategies, including dominant hip strategies and swing initiation alterations or step length modifications. Therefore, a more sophisticated, potentially multimodal analysis of a combination of different parameters and a fusion of data from various body segments, such as the pelvis, thigh, shank, and foot, may offer a better indicator for forward propulsion [15,31]. While the use of multiple IMUs might be feasible for in-clinic rehabilitation, implementing a multi-sensor setup in the home situation in these patients is often unfeasible.

## 5. Conclusions

The findings in the current study offer valuable insights that can contribute to the development of feedback systems aimed at improving the gait pattern of stroke survivors. This study demonstrated that the FSA can be accurately assessed with an IMU on the foot during straight-ahead walking. Our proposed foot and shank movement parameters were not suitable to provide patients with feedback regarding forward propulsion.

## Figures and Tables

**Figure 1 sensors-24-00710-f001:**
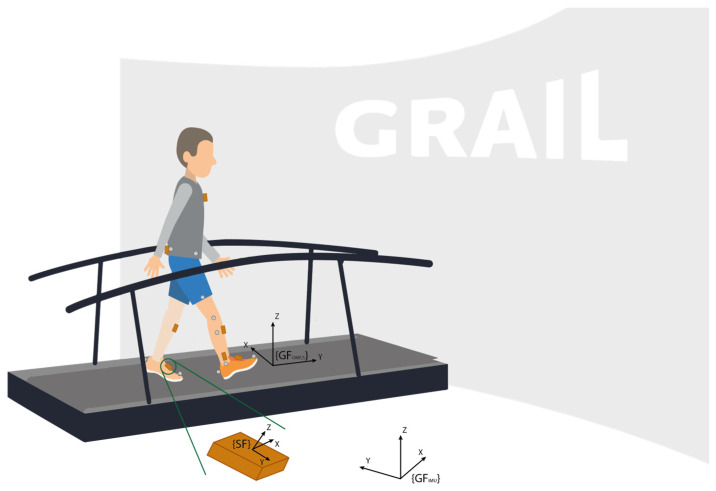
Schematic representation of the measurement setup. Note the grey optical markers at the toe and heel of the feet, defining the foot segment, as well as the markers at the knee and ankle, defining the shank segment. {SF} represents the local sensor frame of the IMU, {GF_OMCS_} represents the global frame of the OMCS system, and {GF_IMU_} represents the global frame of the IMU system.

**Figure 2 sensors-24-00710-f002:**
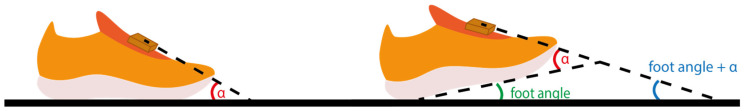
The measured IMU-based foot angle (foot angle + α) corrected with the mean foot angle (α) during the foot flat phase of the first 10 strides, to consider the foot angle during the foot flat phase to be zero degrees.

**Figure 3 sensors-24-00710-f003:**
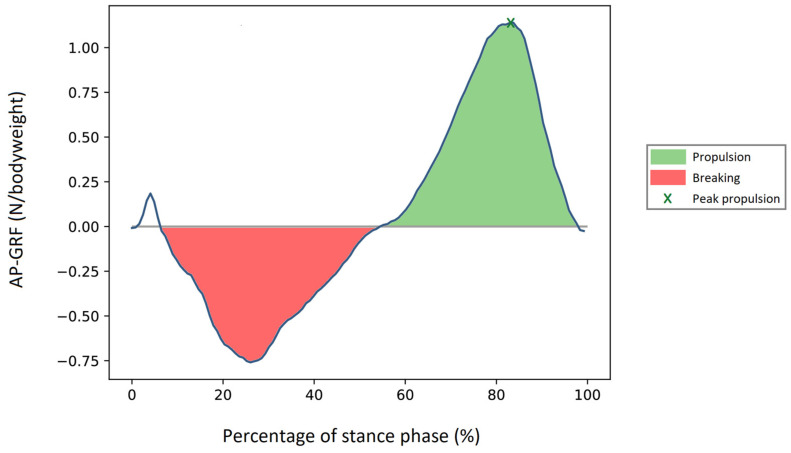
Forward propulsion measured by the area under the curve from the breaking-to-propulsion transition until TC, indicated with green. Peak forward propulsion, indicated by x, was defined as the maximum value from the breaking-to-propulsion transition until TC.

**Figure 4 sensors-24-00710-f004:**
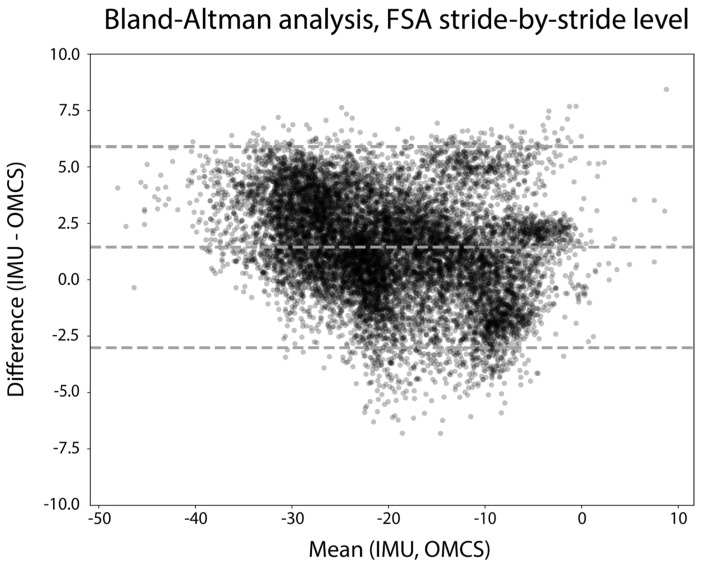
Bland–Altman analysis of the FSA (degrees) of all strides of all participants. The difference between measures is calculated as IMU-based FSA—OMCS-based FSA.

**Figure 5 sensors-24-00710-f005:**
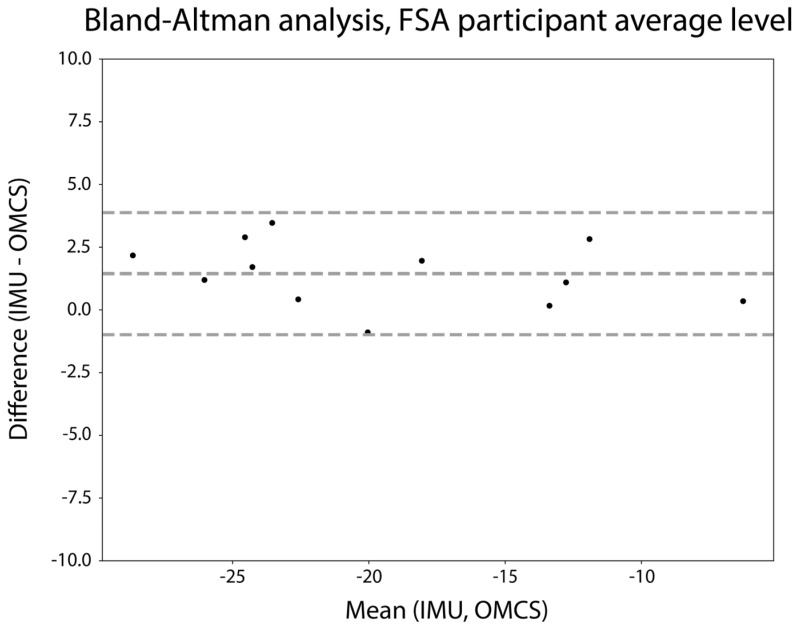
Bland–Altman analysis of the mean FSA (degrees) per participant. The difference between measures is calculated as mean IMU-based FSA—mean OMCS-based FSA.

**Table 1 sensors-24-00710-t001:** Participant characteristics.

Participant Characteristics	
N	12
Gender (male/female)	7/5
Age (mean ± SD years)	61.0 ± 9.5
Height (mean ± SD cm)	176.4 ± 8.5
Weight (mean ± SD kg)	85.0 ± 14.7
Affected side (left/right)	6/6
Stroke type (ischemic/hemorrhagic/unknown)	8/2/2
Time since stroke onset (median (IQR) months)	24.5 (11; 76.5)
Gait speed (mean ± SD m/s)	1.0 ± 0.3

**Table 2 sensors-24-00710-t002:** Pearson correlation between different gait characteristics and the AUC forward propulsion for the stride-by-stride analysis. All parameters are separately evaluated based on OMCS data and IMU data. * indicates significant correlations (*p* < 0.05).

	IMU-Based	OMCS-Based
Parameter	Pearson r	Pearson r
Foot angle upon TC	0.43 *	0.52 *
Max foot angular velocity	0.23 *	0.32 *
Max foot angular acceleration	−0.01	0.18 *
Shank angle upon TC	0.26 *	0.42 *
Max shank angular velocity	−0.13 *	0.12 *
Max shank angular acceleration	0.23 *	0.21 *
Shank linear acceleration	0.17 *	−0.01
Stride length	0.26 *	0.50 *

**Table 3 sensors-24-00710-t003:** Mean and standard deviation of the within-subject analysis for the Pearson correlation between the different gait characteristics and the AUC forward propulsion. All parameters are separately evaluated based on OMCS data and IMU data.

	IMU-Based	OMCS-Based
Parameter	Pearson r Mean ± SD	Pearson r Mean ± SD
Foot angle upon TC	0.44 ± 0.26	0.49 ± 0.31
Max foot angular velocity	0.19 ± 0.37	0.39 ± 0.31
Max foot angular acceleration	0.04 ± 0.26	0.19 ± 0.42
Shank angle upon TC	0.32 ± 0.37	0.63 ± 0.22
Max shank angular velocity	0.01 ± 0.34	0.09 ± 0.37
Max shank angular acceleration	0.17 ± 0.24	0.19 ± 0.42
Shank linear acceleration	0.28 ± 0.17	0.06 ± 0.20
Stride length	0.20 ± 0.26	0.49 ± 0.20

**Table 4 sensors-24-00710-t004:** Pearson correlation between different gait characteristics and the peak forward propulsion for the stride-by-stride analysis. All parameters are separately evaluated based on OMCS data and IMU data. * indicates significant correlations (*p* < 0.05).

	IMU-Based	OMCS-Based
Parameter	Pearson r	Pearson r
Foot angle upon TC	0.61 *	0.77 *
Max foot angular velocity	0.63 *	0.78 *
Max foot angular acceleration	0.05 *	0.64 *
Shank angle upon TC	0.21 *	0.68 *
Max shank angular velocity	−0.14 *	0.53 *
Max shank angular acceleration	0.46 *	0.60 *
Shank linear acceleration	0.38 *	0.35 *
Stride length	0.76 *	0.74 *

**Table 5 sensors-24-00710-t005:** Mean and standard deviation of the within-subject analysis for the Pearson correlation between the different gait characteristics and the peak forward propulsion. All parameters are separately evaluated based on OMCS data and IMU data.

	IMU-Based	OMCS-Based
Parameter	Pearson r Mean ± SD	Pearson r Mean ± SD
Foot angle upon TC	0.47 ± 0.26	0.56 ± 0.37
Max foot angular velocity	0.50 ± 0.22	0.59 ± 0.26
Max foot angular acceleration	0.22 ± 0.17	0.28 ± 0.36
Shank angle upon TC	0.15 ± 0.45	0.55 ± 0.33
Max shank angular velocity	−0.05 ± 0.31	0.16 ± 0.49
Max shank angular acceleration	0.14 ± 0.31	0.28 ± 0.36
Shank linear acceleration	0.25 ± 0.23	0.18 ± 0.18
Stride length	0.20 ± 0.26	0.49 ± 0.20

## Data Availability

All data and code for data processing and analysis are available at https://github.com/SintMaartenskliniek/MovingReality (Release: “Validation study”, tag: “v1.0.0”, assessed on 6 January 2024).

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
