# Peer review of "Assessment of Foot Strike Angle and Forward Propulsion with Wearable Sensors in People with Stroke"

_sensors, 2024, doi:10.3390/s24020710_

Round 1

Reviewer 1 Report

Comments and Suggestions for Authors

This manuscript assesses the validity of IMU-derived FSA on people with stroke compared to the gold standard, and this assessment is conducted on the people with stroke for real-time feedback within home-based training. This research is meaningful.

The suggestion is to add a schematic diagram showing how to get the foot strike angle of the OMCS and the foot angle of the IMU. It would be better to understand the data analysis.

Reviewer 2 Report

Comments and Suggestions for Authors

The paper aims to validate the accuracy of the foot strike angle in individuals with stroke using the inertial measurement unit against the optical motion capture system and to identify IMU-derived parameters indicative of forward propulsion. The research during walking was conducted on 12 stroke survivors equipped with IMUs and markers for optical motion analysis.
The findings were that the foot strike angle can be accurately assessed with an IMU on the foot during straight-ahead walking. The proposed foot and shank movement parameters proved unsuitable for providing feedback regarding forward propulsion during in-clinic therapy.
Strength: A well-written paper with included statistics.
Weaknesses: The method of choice of the eight parameters was not described clearly: previous studies [7,8,12,16–18] (group citation!) used them, but it was not stated (in the introduction) if these parameters are potentially promising in application to the forward propulsion description.

Eq (1): Foot segment Optical Motion Capture System = position TOE – position HEEL,
Comment: I do not understand this equation: Does the system equal the difference between positions (vertical, horizontal)? Could you add a sketch?
196-  Eight parameters were identified ...  Comment: Why just these eight parameters (explanation, relevant citations)? Which parameters were rejected and why? What were the simplifying assumptions? These parameters are scalars in the research. However, they are functions of time. It may be that investigating relationships between their amplitudes and phases in both directions during the gait cycle would supply a more reliable identification of IMU-derived parameters for forward propulsion.
Eq (8): Shank segment Optical Motion Capture System (position?)= position KNEE – position ANKLE,
Comment: See the comment on Eq (1)
Eq (9): Comment: What was the difference between the x-axis and the “walking direction component”
Eq (10): Comment: Do accelerations along both axes have the same phases?

Reviewer 3 Report

Comments and Suggestions for Authors

This study aimed to evaluate IMU-derived Foot Strike Angle (FSA) accuracy and identify indicators for forward propulsion in stroke survivors. Twelve post-stroke participants underwent gait analysis, revealing excellent agreement for FSA. Despite a mean FSA difference of 1.5 degrees, eight potential indicators showed weak to moderate correlations with propulsion. Stride-by-stride and within-subject analyses indicated varied correlations, none reaching a 'good' level for a substantial number of individuals. The study concludes that while IMU accurately assesses FSA, the proposed parameters may not offer meaningful feedback on forward propulsion. It emphasizes the need for refined indicators and acknowledges limitations, providing valuable insights for stroke survivor gait rehabilitation.

Strengths:

  1. The study successfully demonstrates the accurate assessment of Foot Strike Angle (FSA) using an Inertial Measurement Unit (IMU) during straight-ahead walking in stroke survivors. The mean difference of 1.5 degrees and excellent intraclass correlation indicate high accuracy.
  2. The work employs a comprehensive evaluation approach, considering both FSA and potential IMU-derived indicators for forward propulsion. This provides a holistic understanding of gait characteristics in stroke survivors.
  3. The comparison of IMU-derived FSA with a gold standard and the subsequent analysis contribute to the validation of IMU-based measurements, enhancing the reliability of the study's findings.
  4. The within-subject analysis offers insights into individual variability, recognizing the heterogeneity in gait patterns among stroke survivors. This nuanced approach adds depth to the interpretation of results.

Limitations:

  1. The study primarily focuses on straight-ahead treadmill walking, limiting the ecological validity of findings. Real-world scenarios involving curved paths, varied terrains, and uneven surfaces are not adequately addressed.
  2. The heterogeneity in gait patterns among stroke survivors introduces variability, potentially affecting the generalizability of the proposed IMU-derived indicators.
  3. The study concludes that the proposed IMU-derived parameters are not suitable for providing feedback on forward propulsion, highlighting the complexity of identifying meaningful indicators in stroke survivors.
  4. Acknowledging the need for computational efficiency, the study prioritizes relatively simple parameters for forward propulsion, potentially overlooking the benefits of a more sophisticated, multi-segment analysis.
  5. The observed differences in correlation coefficients between IMU-derived and Optical Motion Capture System (OMCS)-derived parameters suggest potential device-specific variations that need further investigation.

There are few points that the authors can work on to enhance the clarity and comprehensibility of their work:

·       Provide a schematic illustration outlining the placement of IMUs and motion analysis markers on the participants during the walking trials.

·       Include a visual representation, such as a graph, comparing the IMU-derived Foot Strike Angle (FSA) with the gold standard (OMCS-based FSA) using Bland-Altman analysis.

·       Present graphs displaying the correlation between various kinematic parameters (foot and shank movement) and forward propulsion, both on a stride-by-stride and within-subject level.

·       Include images or diagrams illustrating variations in gait patterns among stroke survivors, emphasizing the heterogeneity within the study population.

·       Visualize the repeatability coefficients for IMU-based FSA, demonstrating the consistency of measurements on both stride-by-stride and within-subject analyses.

·       Include a photograph showing the placement of IMUs on the foot, providing a clear view of how sensors are attached during the walking trials.

·       Use a visual representation to compare the global coordinate systems of IMUs and Optical Motion Capture System (OMCS), emphasizing any alignment or transformation applied.

·       Display graphs depicting the correlation between kinematic parameters (foot and shank angles, angular velocity, etc.) and forward propulsion, emphasizing the variability among stroke survivors.

·       Provide a visual guide or diagram explaining how gait events (initial contact, terminal contact) were identified using both OMCS and IMU data.

·       Create a simple illustration explaining the adjustment process for foot angle during foot flat phase, clarifying the calculation method mentioned in equations.

-       Please provide the citation next to figure 1 and make sure to get the copyright as well

Comments on the Quality of English Language

The English write-up of the work generally exhibits a clear and technical style. However, there are a few instances where sentence structures could be improved for better readability and flow. Some sentences are complex and may benefit from simplification, while others could be more concise. Overall, the content is well-organized, but minor adjustments in language and structure could contribute to a smoother presentation.

Reviewer 4 Report

Comments and Suggestions for Authors

Assessment of foot strike angle and forward propulsion with 2 wearable sensors in people with stroke

The manuscript is well-written in an engaging and lively style. It’s currently something of a “stormy topic” and is one to which the author has made significant contributions in the validation of a device. The purpose of the manuscript was to validate the accuracy of the IMU-derived FSA in individuals with stroke against the gold standard optical motion capture system (OMCS), and 2) to identify IMU-derived parameters that are indicative of forward propulsion in individuals with stroke. I have some queries regarding the data analysis and representation which can enhance the weightage of the manuscript as I mention step by step below.

L 81, 84, 302, 304, 306, 321… remove personal pronouns.

L 93 Why was the sample size not calculated? Sample size leads to type one error. What do authors do to overcome type one errors?

Which application was run during the walking trials on the GRAIL?

L 218-221 The intraclass correlation coefficient (ICC) is one measure of reliability not validity, and the Bland-Altman plot as a measure of agreement.

Which α level was considered the most significant level?

Indicate the purpose of the p-value, otherwise omit these values.

It will be better if the authors represent the actual p-values.

Why were the mean and SD values for age, length, and weight of the participants in discrete values not in the continuous values?

The word length should be changed with height.

Pearson correlation coefficients are represented by the 'r' not the "ρ".

Mention the units for each parameter in tables.

Table 2, 4, & 5 - why is Max shank angular velocity in negative values for IMU-based?

Table 3, 5 remove the Pearson ρ from the cell if it represents only mean values.

Kindly follow similar style to represent mean and SD as table 1 (mean ± SD) and table 3 (Mean       SD).

Round 2

Reviewer 3 Report

Comments and Suggestions for Authors

I appreciate the efforts the authors have put into addressing the suggested improvements and making necessary corrections to the manuscript.

The additions of schematic illustrations, Bland-Altman plots, correlation graphs, and other visual representations have significantly enhanced the clarity of the study. The authors' detailed explanations and the inclusion of Figures 1 and 2 provide a clearer understanding of the measurement setup and adjustment processes, contributing to the overall quality of the manuscript.

I'm pleased to see the authors' commitment to transparency, particularly in acknowledging and correcting the typo in the data-analysis script. The authors' attention to detail and responsiveness to the feedback are commendable.

I believe these revisions have strengthened the manuscript, and I look forward to the final version.

Comments on the Quality of English Language

The English write-up of the work generally exhibits a clear and technical style. 

Reviewer 4 Report

Comments and Suggestions for Authors

All the queries are well answered by the authors, and the due changes have been made. The manuscript should move further for acceptance. 

Authors should provide parameter units in tables 3 & 5, there are p or r values, only mean and SD. 
